# Evaluation of Anti-HB Levels in a Multi-Ethnic Cohort of Health Profession Students [note 1]

**DOI:** 10.3390/vaccines13070771

**Published:** 2025-07-21

**Authors:** Lorenzo Ippoliti, Andrea Pizzo, Agostino Paolino, Luca Coppeta, Giuseppe Bizzarro, Cristiana Ferrari, Andrea Mazza, Claudia Salvi, Ersilia Buonomo, Fabian Cenko, Andrea Magrini, Antonio Pietroiusti

**Affiliations:** 1Department of Biomedicine and Prevention, University of Rome Tor Vergata, 00133 Rome, Italy; 2PhD Program in Social, Occupational and Medico-Legal Sciences, Department of Occupational Medicine, University of Rome Tor Vergata, 00133 Rome, Italy; 3Faculty of Medicine, Saint Camillus International University of Health Sciences, 00131 Rome, Italy; 4Faculty of Medicine, University “Our Lady of Good Counsel”, 1000 Tirana, Albania

**Keywords:** hepatitis B, vaccination, occupational medicine, healthcare workers

## Abstract

Background: Despite the widespread implementation of childhood vaccination programmes, hepatitis B virus (HBV) infection remains an ongoing occupational risk for healthcare students. In multi-ethnic and international university settings, differences in vaccination programmes and immune responses must be considered. This retrospective study aimed to assess the prevalence of protective levels of anti-HBs among medical students at an international university in Rome, exploring associations with demographic and vaccination-related factors. Methods: Data were collected from routine occupational health surveillance conducted in 2023. Anti-HB titres were measured in 507 students, and information on age, sex, country of birth, age at vaccination, and time since the last dose was analysed. Results: Overall, 55.0% of students had antibody levels of at least 10 mIU/mL, indicating serological protection. Higher seroprotection rates were observed among students vaccinated in the first year of life compared to those vaccinated later. A significant decline in antibody titres was also associated with longer intervals since vaccination. Students born outside Europe tended to show lower levels of protection. Conclusions: These results emphasise the importance of screening future healthcare professionals and continuously monitoring antibody titres to help reduce HBV infections.

## 1. Introduction

The prevalence of hepatitis B virus (HBV) in the European Union is estimated at 0.9%, corresponding to approximately 4.7 million chronic cases. Notably, this figure reflects a sustained decline over the past three decades [1]. This downward trend is not limited to Europe but is part of a broader global reduction in HBV prevalence [2,3], attributable to improved prevention, vaccination, and public health strategies. However, on a global scale, HBV remains a major health concern, with an estimated 260 million people living with chronic infection [4,5]. The hepatitis B virus is primarily transmitted through contact with infected blood or other bodily fluids [6]. In high-endemic areas, perinatal transmission—where the virus is passed from mother to child at birth—is also a common mode of infection [7].

The course of HBV infection is heavily influenced by the host’s immune response and the presence of any coexisting conditions that may affect immunological function [8]. Depending on these factors, the primary infection may either resolve spontaneously or evolve into a chronic condition. Generally, 5–10% of acute infections acquired during adulthood progress to chronic hepatitis B, whereas the risk rises significantly with earlier age at infection, reaching up to 90% in cases of perinatal acquisition [9].

Chronic HBV infection follows a complex clinical trajectory. Over time, it may lead to progressive liver damage, including fibrosis and cirrhosis, and eventually hepatocellular carcinoma, the most common form of primary liver cancer [10]. The burden of chronic HBV is therefore not limited to liver-related morbidity, but extends to substantial healthcare costs and loss of productivity, especially in low-resource settings where access to diagnostics and antiviral therapies may be limited.

Among high-risk groups, healthcare workers (HCWs) and students enrolled in health-related academic programs are particularly vulnerable to HBV infection due to frequent exposure to blood and other potentially infectious materials [11]. Students are considered especially at risk during internship periods, where their limited experience and inconsistent adherence to safety protocols may contribute to increased exposure [12]; moreover, educational programs on occupational risks and preventive measures are sometimes inadequate, further compounding their vulnerability [13]. As a result, medical and healthcare students are usually subject to regular occupational health assessments prior to clinical placements. These include immunization reviews and serological evaluations to ensure adequate protection against transmissible diseases, particularly hepatitis B.

In most industrialized countries, including Italy, HBV vaccination has been included in the national immunization schedule for decades. The three-dose vaccine regimen is typically initiated in infancy, with the first dose administered within the first year of life [14]. Nevertheless, not all individuals receive the full series in childhood. Incomplete vaccination may result from various factors, including neonatal health issues, vaccine shortages, or vaccine hesitancy and parental refusal. Consequently, there remains a subset of young adults who require catch-up vaccination or additional serological monitoring to ensure protection.

One of the key assessments during health surveillance is the measurement of hepatitis B surface antibodies (anti-HBs), which are indicative of protective immunity. These antibodies can develop as a result of either successful vaccination or recovery from a resolved HBV infection [15]. Although anti-HBs are not the only antibodies associated with HBV—others, such as anti-HBc, may reflect prior exposure—the presence of anti-HBs in isolation typically confirms immunization without active or past infection [16].

Scientific evidence indicates that an anti-HB titre of ≥10 IU/L is considered the protective threshold, ensuring effective immunity against HBV for at least 10 years following the last vaccine dose [17]. In many cases, protection may persist for up to 30 years, even in individuals whose antibody levels decline below detectable limits [18]. However, in occupational settings where HBV exposure risk is high, such as hospitals and clinical laboratories, it is common practice to administer a booster (fourth) dose to individuals with anti-HB levels below 10 IU/L, followed by reassessment to confirm an adequate immune response.

Maintaining sufficient antibody levels among healthcare students is not merely a precautionary measure—it is a fundamental component of occupational safety. Numerous studies have reported cases of HBV infection in students and young HCWs, some of which have occurred during internships or early career stages [19,20,21,22]. These cases emphasize the vulnerability of this demographic and underscore the necessity of proactive immunization monitoring.

While HBV vaccination is not legally mandatory in many countries for clinical internships or employment in healthcare settings, it remains strongly recommended and is often de facto required by institutions to minimize liability and ensure patient and staff safety [23]. Nevertheless, the implementation of preventive measures varies across institutions and regions. In some cases, a lack of standardized protocols or inconsistencies in record-keeping can hinder effective follow-up and re-vaccination strategies [24,25,26].

Furthermore, disparities in antibody levels have been observed across populations, suggesting that host factors—such as age at vaccination, time elapsed since the last dose, genetic background, and immune status—can influence long-term seroprotection [18,27]. International students or those from regions with lower vaccination coverage may require closer evaluations to align with local health and safety standards.

In this context, a thorough assessment of anti-HB titres among students in health-related degree programmes is of particular relevance. Such evaluations not only provide insight into the immunological status of future healthcare workers, but also serve as a proxy for the effectiveness of national vaccination strategies and institutional safety protocols [28]. Monitoring antibody titres can guide public health interventions, identify at-risk individuals, and ultimately contribute to reducing the incidence of occupational HBV transmission.

This study was designed to assess the anti-HB titres in a multi-ethnic population of healthcare students, with the aim of identifying the proportion of individuals who are adequately protected, those who may benefit from booster doses, and any potential disparities based on vaccination timing or origin. By focusing on a diverse student cohort, this research also aims to highlight the challenges and opportunities in harmonizing vaccination practices in an increasingly globalized educational environment.

This study aims to evaluate the variables that may influence anti-HB titres after vaccination in a multi-ethnic population of healthcare students at an international university. To this end, data from the annual occupational health surveillance, collected during occupational medicine visits, were analysed.

## 2. Material and Methods

This study analysed data collected during the occupational health surveillance of students enrolled on various healthcare degree programmes (e.g., Medicine, Dentistry, and Nursing) at our university. This surveillance formed part of the mandatory pre-enrolment medical examination conducted between January and September 2023.

Only students with documented evidence of having received all three doses of the hepatitis B vaccine, including the precise dates of administration, were included in the analysis. Students with incomplete vaccination records or missing documentation regarding vaccine dates or serological testing were excluded. In accordance with institutional occupational health protocols, all students undergo hepatitis B surface antigen (HBsAg) screening prior to clinical internship placement. All participants in this study were confirmed to be HBsAg-negative during health surveillance. Data were obtained from the occupational health and risk assessment file compiled during the health surveillance visit. The variables considered in this study were sex, age, country of birth, age at the time of the first vaccine dose, the time elapsed between the last vaccine dose and the serological test, and the presence of a protective antibody response (defined as anti-HBs >10 IU/L). Regarding the variable of the age at which the first vaccine dose was administered, the cut-off point of 12 months was selected to reflect the standard national immunization guidelines, which recommend completing the hepatitis B vaccine series within the first year of life. To explore the relationship between the antibody response and the collected variables, descriptive analyses of the sample were first conducted. For the statistical analysis, categorical variables were compared using the chi-square test. Subsequently, we performed a logistic regression analysis to identify potential predictors of a protective antibody response. Additionally, a chi-square analysis was conducted, stratified by students’ region of birth, to assess potential differences in protective antibody titres across geographic origins. A *p*-value of less than 0.05 was considered statistically significant.

## 3. Results

Out of the total screened student population (*n* = 562), 55 subjects were excluded due to incomplete vaccination documentation or missing serological data. Therefore, 507 students were included in the final analysis, corresponding to an estimated complete vaccination rate of approximately 89%. The characteristics of the study population are summarized in Table 1. The majority of participants were female (64.3%, *n* = 326). The mean age of the population was 23.9 ± 4.3 years (range 18–44 years). The majority of students (71.2%, *n* = 364) were born in Italy and received their hepatitis B vaccination there, while 28.2% (*n* = 143) were born abroad. The latter group included individuals from Europe (0.5%), the Americas (5.0%), Africa (6.1%), and Asia (16.6%).

Students were enrolled in various healthcare programmes: Midwifery (30.6%, *n* = 155); Nursing Sciences (29.6%, *n* = 149); Physiotherapy (18.5%, *n* = 94); Medicine and Dentistry (13.8%, *n* = 70); Biomedical Laboratory Techniques (4.5%, *n* = 23); and Medical Radiology Techniques (3.0%, *n* = 15).

In terms of vaccination timing, 47.5% (*n* = 241) of students had completed the hepatitis B vaccination series within the first 12 months of life, 32.7% (*n* = 166) had received it between 12 and 24 months of age, and 19.8% (*n* = 100) had received it after the age of two. The median age at vaccination was 1 year (IQR: 0–1), with a mean of 4.4 ± 9.6 years (range: 0–43), indicating a right-skewed distribution. The mean age at antibody testing was 21.9 ± 4.6 years (range 17–44). The average interval between the last vaccine dose and antibody testing was 17.4 ± 6.9 years (range 0–30 years).

A protective antibody titre (>10 IU/L) was observed in 55.0% of the sample (*n* = 279).

For the statistical analysis, continuous variables were categorized using the chi-square test. Specifically, age at the time of testing was divided based on the median value: ≤23 years (61.1%, *n* = 310) and >23 years (38.9%, *n* = 197); age at completion of the vaccination cycle was grouped as follows: ≤12 months (47.5%, *n* = 241) and >12 months (52.5%, *n* = 266); the time elapsed since the last vaccine dose was categorized as ≤20 years (75.1%, *n* = 381) or >20 years (24.9%, *n* = 126).

The chi-square analysis (see Table 2) revealed significant associations between a protective antibody response (defined as anti-HBs >10 IU/L) and several variables. Females had a significantly higher proportion of protective titres (37.7%) than males (17.4%) (*p* < 0.05). Participants aged 23 years or under had higher seroprotection rates (38.5%) than those aged over 23 years (16.6%) (*p* < 0.05). The proportion of students born in Italy with protective titres was significantly higher than that of students born abroad (43.6% vs. 11.4%, *p* < 0.05). Those vaccinated within the first year of life had a higher prevalence of protective titres (28.8%) than those vaccinated later (26.2%) (*p* < 0.05). A longer time interval since the last vaccination was associated with a weaker antibody response. A total of 45.0% of students who were vaccinated 20 years ago or less had a protective titre, compared to 10.1% of those who were vaccinated more than 20 years ago (*p* < 0.05).

The logistic regression analysis (Table 3) confirmed that being born outside Italy was significantly associated with a lower likelihood of seroprotection (OR = 0.45; 95% CI: 0.29–0.71; *p* < 0.05). Similarly, being vaccinated after the first year of life (OR = 0.66; 95% CI: 0.44–0.99; *p* < 0.05) and having an interval of more than 20 years between vaccination and antibody testing (OR = 0.39; 95% CI: 0.23–0.67; *p* < 0.05) were independently associated with a lower likelihood of a protective antibody response. Gender and age at the time of testing were not statistically significant predictors in the regression model. The logistic regression model demonstrated an acceptable fit, with a Nagelkerke pseudo R^2^ value of 0.11.

A further chi-square analysis, stratified by birth region (Table 4), indicated that the highest proportion of protective titres was found among students born in Italy or other European countries (44.0%). Lower seroprotection rates were observed in students from Asia (6.5%), Africa (2.2%), and the Americas (2.4%) (*p* < 0.05).

## 4. Discussion

The findings of this study highlight a significant public health concern: a large proportion of healthcare students who were screened had non-protective anti-HB titres. This raises concerns about the long-term persistence of immunity following hepatitis B vaccination. These results emphasise the multifactorial nature of vaccine-induced immunity, suggesting that various demographic and clinical factors can affect the longevity of seroprotection. In line with previous research, female students demonstrated significantly higher levels of seroprotection than their male counterparts [27,29]. This finding is consistent with studies on other vaccinations [30] and may partly be attributed to biological differences in immune function. It is well established that females generally mount stronger humoral responses, potentially due to hormonal and genetic factors influencing immune regulation.

Age also emerged as a relevant variable, with younger students being more likely to retain protective antibody levels. This finding corroborates the well-recognised phenomenon of waning immunity over time [31]. Notably, the time elapsed since the initial vaccination series was a critical determinant of serological status: students vaccinated over 20 years prior to testing exhibited significantly lower seroprotection rates. This finding is consistent with previous evidence indicating that, while hepatitis B vaccination is highly effective initially, antibody levels tend to decline in the absence of booster doses or natural antigenic exposure [27,30].

Interestingly, students who were vaccinated within the first year of life were more likely to maintain protective titres. This could reflect more effective immunological priming during early infancy, as well as greater adherence to standardized national immunization schedules. However, the literature presents mixed findings regarding this association, with some studies failing to confirm a long-term protective effect linked to early vaccination timing [27,30,32]. These inconsistencies highlight the complexity of vaccine response dynamics and emphasise the need for further longitudinal studies.

While age and sex were statistically significant in univariate analyses, they did not retain significance in the multivariate logistic regression model. This suggests that their apparent effects may be mediated or confounded by other variables, most notably the timing of vaccination and country of birth.

One of the most striking findings from this study relates to the geographic origin of the students. Those born outside Italy, particularly in Asia and Africa, had significantly lower seroprotection rates. The stratified chi-square analysis further substantiated this trend, identifying the lowest antibody levels among students from these regions. Such disparities likely reflect heterogeneities in global vaccination policies, including variations in the age at which vaccines are administered, coverage rates, storage and handling conditions, and access to healthcare services. Furthermore, broader socioeconomic and health system inequalities in the countries of origin may contribute to these immunization disparities.

In the context of increasing international mobility and the diversification of student populations within healthcare education programmes, these findings carry important implications. They reinforce the critical need for the comprehensive screening of all healthcare students upon entry to university—not only to assess immunity to hepatitis B, but also as a model for evaluating protection against other vaccine-preventable diseases. In this era of immunization, ensuring uniform protection among future healthcare professionals is a public health priority. Failure to identify and address immunity gaps could undermine infection control efforts, posing a risk to patients and clinical staff alike.

Taken together, these observations suggest that, while universal screening is essential, particular attention should be given to students who were vaccinated more than two decades ago and to those born in regions with lower vaccine coverage or inconsistent immunization practices. Tailored booster vaccination strategies informed by serological screening may be a prudent measure to safeguard individual and public health in clinical settings.

This study presents certain limitations that warrant consideration. First, although the cohort was relatively large and included students from multiple continents, the analysis was conducted within a single international university. As such, the findings may not be fully generalizable to other institutions or educational settings with differing vaccination policies or student demographics. Second, 55 students were excluded from the analysis due to incomplete documentation of vaccination history or missing serological data. While this methodological choice was essential to ensure the validity of the findings, it may have introduced a selection bias. Nonetheless, based on the number of exclusions, we estimate a complete hepatitis B vaccination rate of approximately 89% within the screened population—slightly higher than the global average of 85% reported by recent estimates [33].

Additionally, the retrospective nature of this study limited the availability of information on potentially influential variables such as nutritional status, coexisting medical conditions, or adherence to immunization schedules in early childhood. These factors may contribute to interindividual variability in immune response but could not be accounted for in our analysis. Finally, the cross-sectional design of this study precludes the establishment of causal relationships between the variables examined and serological protection. Longitudinal studies would be required to assess the persistence of anti-HB titres over time and the long-term effectiveness of booster vaccination strategies.

## 5. Conclusions

Antibody screening for hepatitis B should be considered a fundamental preventive measure for all healthcare students, given their future occupational exposure risk. This study highlights that factors such as time since vaccination, age at vaccination, and country of birth significantly influence the persistence of protective anti-HB titres. While universal screening is recommended, particular attention should be given to students born outside Italy and to those vaccinated more than 20 years prior to testing, as these groups are more likely to have non-protective antibody levels. These findings support the implementation of targeted booster vaccination strategies for these at-risk subgroups.

## Figures and Tables

**Table 1 vaccines-13-00771-t001:** Characteristics of study population.

		*n*	%	Mean (S.D.)
Total		507		
Age				23.9 (4.3)
Gender	Male	181	35.7	
	Female	326	64.3	
Birth Country	Italy	364	71.2	
	Abroad	143	28.2	
Age at vaccination				4.4 (9.6)
	<12 months	241	47.5	
	12–24 months	166	37.2	
	>24 months	100	19.8	
Age at antibody testing				21.9 (4.6)
	<20 years	381	75.1	
	>20 years	126	24.9	
Anti-HB level	<10 IU/L	228	45.0	
	>10 IU/L	279	55.0	

**Table 2 vaccines-13-00771-t002:** Association between demographic and vaccination-related variables and anti-HB titres stratified by serological protection status.

Anti-HBs						
		−	%	+	%	*p* Value
Gender	Male	93	18.3	88	17.4	<0.05
	Female	135	26.6	191	37.7	
Age	≤23 years	115	22.7	195	38.5	<0.05
	>23 years	113	22.3	84	16.6	
Birth Country	Italy	143	28.2	221	43.6	<0.05
	Abroad	85	16.8	58	11.4	
Age at vaccination	≤1 year	95	18.7	146	28.8	<0.05
	>1 year	133	26.2	133	26.2	
Distance vaccination—test	≤20 years	153	30.2	228	45.0	<0.05
	>20 years	75	14.8	51	10.1	

**Table 3 vaccines-13-00771-t003:** Logistic regression.

	B	S.E.	*p* Value	OR	CI 95%
Gender (Male)	−0.17	0.20	Ns	0.84	0.57–1.24
Age (>23 years)	−0.20	0.24	Ns	0.81	0.51–1.30
Birth Country	−0.80	0.23	<0.05	0.45	0.29–0.71
Age at vaccination (<1 years)	−0.41	0.21	<0.05	0.66	0.44–0.99
Distance vaccination—test	−0.94	0.27	<0.05	0.39	0.23–0.67

**Table 4 vaccines-13-00771-t004:** Chi-square test–birth region.

Anti-HBs					
	−	%	+	%	*p* Value
Italy/Europe	144	28.4	223	44.0	<0.05
America	13	2.6	12	2.4	
Asia	51	10.1	33	6.5	
Africa	20	3.9	11	2.2	

## Data Availability

Datasets used and/or analysed during the current study are available from the corresponding author on reasonable request.

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
