# Peer review of "Evaluation of Anti-HB Levels in a Multi-Ethnic Cohort of Health Profession Students"

_vaccines, 2025, doi:10.3390/vaccines13070771_

Round 1

Reviewer 1 Report

Comments and Suggestions for Authors

Interesting manuscript, authors should mention the exclusion rate as this will reflect the real world vaccination rate

In statistics- continuous variables were converted to categorical , rational for this not mentioned

Statistics for log regression can be improved by mentioning about "fitness model: ie value of Pseudo R square value

The references need to be amended

Authors should mention if ethics committee approval was taken for research on humans, if approval waived was this based on approved audit or other regulatory approval conditions ?

Reviewer 2 Report

Comments and Suggestions for Authors

This is a well-written study of the presence of seroprotective anti-HBs antibody levels in students in healthcare associated programs. Variations in seroprotection, and assumptions about implications of discrepancies with access to hepatitis B vaccination are described. Overall there was a surprisingly low rate of seroprotection in this relatively young cohort. My main concern is that the students that were born in regions that have high endemicity of chronic HBV infection were the ones most likely to have no HBV seroprotection. Are these students also tested for HBsAg? If not, how can the authors ensure that they are not actually detecting students with chronic HBV infection? The health implications of this possible outcome are also important. If this was not done, then this lapse needs to be explained in study limitations.

Comments and questions:

 Page 4: The mean age at vaccination was 4.4 years. However, 80% of students had been vaccinated before age 24 months and the SD was huge, consistent with high skewness/kurtosis. Consider median and IQR to describe the distribution of this variable.

Page 4: Age at time of testing was split by median value, but age at time of completion of vaccine was noted to be 12 months without explanation. Was this also a median value?

Page 4 – 5: Table 1 detailed 507 students but Table 2 only had 228 students. I could not find an explanation for the missing students. The reasons for the missing data could greatly impact interpretation of these results.

Page 5: Exp(B) is more commonly described as an odds ratio.

Round 2

Reviewer 2 Report

Comments and Suggestions for Authors

The authors have adequately addressed raised concerns. 

It looks like typographical errors have been introduced during copyediting (see instances of "immunization") but they look deliberate and I don't understand them. Suggest carefully going through copy and ensuring adherence to style standards (for example, anti-HBs is referred to several different ways in this manuscript). Otherwise I think this is an important study and will contribute to our HBV elimination data.  

Author Response

We thank the Reviewer for their positive feedback and for highlighting the typographical inconsistencies. We apologize for the errors introduced during the copyediting process and have carefully reviewed the manuscript to ensure consistency in terminology and overall adherence to style standards.